# OpenReview forum: "Evaluating Large Language Models through Role-Guide and Self-Reflection: A Comparative Study"
_ICLR.cc/2025/Conference — ICLR 2025 Poster_

### Official Review · Reviewer_fznJ · 2024-10-20

**Soundness:** 2
**Presentation:** 2
**Contribution:** 2
**Rating:** 6
**Confidence:** 3

**Summary:**

This paper proposed RoSe, which is a set of strategies for assessing whether LLMs truly know the world knowledge, and how their confidence in their prediction could be affected when their answers are challenged by different roles. The authors also propose a double-calibrated strategy to fine-tune open-source LLMs so that they are more robust to local misleading information.

**Strengths:**

The studied question may be of great importance for the community to know the essence of LLMs and to develop better models. It is interesting to find how different types of guidance/challenges could affect the LLM results. The authors have made some effort to support their claim with experimental evidence.

**Weaknesses:**

- Many existing research articles studied the question of "whether LLMs truly know what they know". Although this article attends to more specific aspects of when and how LLM could fail, the demonstrated results are intuitive and may not deserve the discussion using a 10-page conference paper.

- The narration and illustration could use some improvement. For example, Figure 2 is not so informative in presenting the RoSe strategy or how the dataset for calibrated fine-tuning is constructed. There are significant redundancies within the first and second paragraphs in Section 4.2. The concept of "well-calibrated data" is not well-introduced and should be discussed in detail as it plays a key role in the fine-tuning process, etc.

- Some choices are not fully explained. For example, why the authors choose to do the main evaluation on the self-developed EG-QA dataset rather than other open-source datasets such as BBH, which also provides CoT chains in their answers.

- The reproducibility might be an issue. The proposed dataset EG-QA is not shared, the GPT versions are not specified, the fine-tuning objective is not sufficiently elaborated, etc.

**Questions:**

Edit 11/21: fix typos in the original comments.

- The narration in Lines 180--182 is confusing. What does it mean by we can obtain *, satisfying * based on logical consistency? Why the terms p and q are removed from $p(a,c|r)$? Does it mean the answer and confidence are generated only based on the reasoning chain, without seeing the original prompts?

- The results tables (2,3,4,5) show poor model calibration.

- How was the verbal confidence level such as "very confident" converted to scores?

- This paper is based on the assumption "students who don’t really know are easily affected by the teacher and peer guidance". Is there any evidence proving that this also holds for LLMs? This paper shows that LLMs are affected to different degrees by different types of guidance, but it does not directly build the link between "not really know" and "easily affected".

---

> ### Author Response · Authors · 2024-11-20
> **Response to Reviewer fznJ (1/4)**
>
> We sincerely appreciate your valuable time and feedback on our paper and are pleased to know that you recognize the great importance of the community and our interesting findings! We are committed to thoroughly addressing your concerns.
>
> ---
>
> **Comment1**: Many existing research articles studied the question of "whether LLMs truly know what they know". Although this article attends to more specific aspects of when and how LLM could fail, the demonstrated results are intuitive and may not deserve the discussion using a 10-page conference paper.
>
> **Answer**: Thanks for the thoughtful question! Although existing works proposed evaluation work from different aspects, we approach the evaluation in a more fine-grained, specific and novel way, the findings and conclusions we draw need work to prove with experimental results. Meanwhile, we propose a double-calibrated strategy to automatically obtain high-quality reasoning data and fine-tune open-source LLMs to improve their reasoning capabilities.
>
> We reiterate our findings and work on evaluation and fine-tuning here. In the evaluations on closed-source LLMs, which can be found in Section 5.4.1:
>
> > 1. LLMs tend to capture shortcuts by relying solely on the strong reminder “answer is” in prompts to quickly find the answer rather than understanding genuine relationships between prompt and truth during training;
>
> > 2. Under the guidance of error information, LLMs fail to adhere to their own correct answer, exhibiting uncertainty on themselves;
>
> > 3. LLMs tend to trust the role of authority more;
>
> > 4. The overall confidence level of LLMs in settings with random cues is lower than that in settings with truth cues. &  The overall confidence of the LLMs at step-3 decreases under the guidance of different roles compared to the no-role guidance, which is similar to student performance and reflects their uncertainty.
>
> In the fine-tuning of open-source LLMs, which can be found in  Section 5.4.2:
>
> Through the conclusions obtained in the evaluation stage: LLMs rely less on strong reminder information under role guidance (line 377-411), while LLMs are able to enhance their reasoning process during self-reflection. We regard the data that reason correctly and gain confidence in the reasoning process as well-calibrated data, which can help enhance the reasoning ability of the open-source LLMs and alleviate their shortcut learning by finding answers through prompts. Extensive experiments on the ID, OOD sets of EG-QA and openBookQA datasets demonstrate the effectiveness of the strategy.
>
> ---
>
> **Comment2**: The narration and illustration could use some improvement. For example, Figure 2 is not so informative in presenting the RoSe strategy or how the dataset for calibrated fine-tuning is constructed. There are significant redundancies within the first and second paragraphs in Section 4.2. The concept of "well-calibrated data" is not well-introduced and should be discussed in detail as it plays a key role in the fine-tuning process, etc.
>
> **Answer**: Thanks for the detailed suggestion! We have modified Section 4.2 in the updated version, and we mainly illustrate the RoSe strategy through the case in Figure 1, which contains the role guidance, and three-step reflection strategy using prompt settings. We also answer specifically to the reviewer's question here and explain them in the updated version.
>
> - In lines 194-197, the **RoSe strategy** is implemented through three main steps: In the first step, the LLM generates an initial response. In the second step, the model engages in reflective analysis of its previous answer. Finally, the model receives guidance from relevant roles, which offers a reference answer, enabling the model to further reflect on its prior response while integrating role guidance.
>
> - The main goal of **calibrated fine-tuning** is to obtain advanced reasoning data from closed-source LLM and fine-tune open-source LLMs to improve their reasoning capabilities. By leveraging the strong reasoning power of closed-source LLM, we extract well-calibrated data without human annotations.
>
> - In line 234-240 (in revision line 230-234), **well-calibrated data** refers to data that ensure both the accuracy and authenticity of the reasoning process (LLM truly knows how to solve it). It maintains the model's confidence throughout, preventing a significant drop in confidence levels at the final step compared to previous ones. This mitigates the effect of reminder in role guidance that could undermine the model’s certainty (in line 423-426/revision line 428-430). Such well-calibrated data ensures not only the high quality but also the consistency of the reasoning process.
>
> ---

---

> ### Author Response · Authors · 2024-11-20
> **Response to Reviewer fznJ (2/4)**
>
> **Comment3**: Some choices are not fully explained. For example, why the authors choose to do the main evaluation on the self-developed EG-QA dataset rather than other open-source datasets such as BBH, which also provides CoT chains in their answers.
>
> **Answer**: Thanks for your insightful question! As we mentioned in Section 5.1, the EG-QA dataset primarily consists of standardized multiple-choice examination questions collected from [website](http://www.zxxk.com/), which is an authoritative educational resources website, designed for Chinese teachers and students to provide teaching resources, learning resources, providing examination papers, teaching courseware and other text materials. The data in EG-QA include questions, standard answers, and knowledge points, requiring no additional manual annotation. The data originates from after December 2023, ensuring high quality and no data contamination issues. EG-QA is currently available [here](https://anonymous.4open.science/r/EG-QA-C2B2).
>
> Inspired by findings in educational psychology, we adopt exam questions to evaluate LLMs under RoSe strategy. To ensure high stability, consistency, and reproducibility in evaluations, the options (letters) of multiple-choice QA tasks serve as cue information in the RoSe strategy without deviating from the question itself, ensuring stable assessments. Although there are several open-source multiple-choice datasets (BBH [1], RACE [2], MT-test [3], etc.), they fail to meet some requirements of the paper:
>
> - **Test Environment for Fine-Tuning LLMs**:
> In this paper, we propose a double-calibrated strategy to effectively fine-tune open-source LLMs. Since the fine-tuning process is prone to overfitting, it is necessary to conduct a full evaluation of both ID and OOD datasets. Current QA datasets often cannot guarantee they weren't employed during pre-training or RLHF processes (data contamination issue commented by Reviewer nbt9). They also lack clear ID and OOD divisions, hindering the assessment of generalization. The EG-QA dataset addresses these issues with newly collected data and contains diverse knowledge points, providing clear ID and OOD splits. In addition, fine-tuning an LLM might affect the LLM's ability in commonsense reasoning, and we also employ open-source openBookQA for detailed evaluation.
>
> - **Bilingual Multiple-Choice QA Dataset**:
> EG-QA is designed for Chinese students and includes both English and Chinese content, making it a bilingual dataset that better assesses model performance. For example, we discovered that LLaMA3-8B still faces challenges with bilingual questions. The questions require models to comprehend and integrate multiple information sources, enhancing their reasoning and application of internal knowledge. Therefore, EG-QA provides a valuable open-source project for the NLP community and researchers needing English multiple-choice QA datasets.
>
> - **No Manual Annotation Required**:
> EG-QA is entirely based on real exam questions from students across various regions in China, accurately reflecting the educational scenarios, which avoids the influence of human annotation preference (issue commented by Reviewer nbt9).
>
> [1] Suzgun M, Scales N, Schärli N, et al. Challenging BIG-Bench Tasks and Whether Chain-of-Thought Can Solve Them, Findings of ACL, 2023.
>
> [2] Lai G, Xie Q, Liu H, et al. RACE: Large-scale ReAding Comprehension Dataset From Examinations, EMNLP 2017: 785-794.
>
> [3] Hendrycks D, Burns C, Basart S, et al. Measuring Massive Multitask Language Understanding, ICLR 2021.
>
> ---
>
> **Comment4**: The reproducibility might be an issue. The proposed dataset EG-QA is not shared, the GPT versions are not specified, the fine-tuning objective is not sufficiently elaborated, etc.
>
> **Answer**: Thanks for your question! We will open-source our code, and dataset, EG-QA is currently available [here](https://anonymous.4open.science/r/EG-QA-C2B2). In line 798 (revision line 800), we mention that the version of GPT-4 is the latest GPT-4 turbo-0409. In addition, we highlight GPT-3.5 in the updated version as GPT-3.5 turbo-1106.
>
> In the answer to Comment2, we highlight the purpose of fine-tuning. Besides, open-source LLMs often have weaker reasoning and instruction-following abilities compared to closed-source LLMs. Fine-tuning aims to enhance these reasoning skills while focusing less on strong reminder information, as demonstrated in Appendix B.3.
>
> ---

---

> ### Author Response · Authors · 2024-11-20
> **Response to Reviewer fznJ (3/4)**
>
> **Comment5**: The narration in Lines 180--182 is confusing. Does it mean the answer and confidence are generated only based on the reasoning chain, without seeing the original prompts?
>
> **Answer**: Thanks for the great question! Answers and confidence levels are generated incrementally. Although the overall process is still dependent on the original question $q$ and prompt $\wp$, the model is updated after each reasoning step based on the current reasoning result. Therefore, the final answer and confidence are always associated with the original $q$ and $\wp$. As illustrated in the paper:
>
> >We want to maximize the conditional probability of $r$, $a$, $c$: $P(y|\wp,q) = P(r,a,c|\wp,q)$. Based on logical consistency $P(a,c|r)$, we could obtain  $P(r|\wp,q) \cdot P(a,c|r,\wp,q)$, signifying $P(r|\wp,q) \cdot P(a,c|r)$.
>
> Logical consistency here does not mean discarding $q$ and $\wp$ completely but refers to the internal self-consistency of LLM, that is, the reasoning and the final answer at each step in the reasoning analysis $r$ should be logical, and the confidence $c$ should be consistent with the correctness of the answer $a$. Thus, although the reasoning analysis can be generated independently, it still relies on $q$ and $\wp$ to maintain a suitable reasoning path and ensure the soundness of the chain of thought.
>
> ---
>
> **Comment6**: The results tables (2,3,4,5) show poor model calibration. In fact, the verbal confidence scores
>
> **Answer**: Thanks for your question! Actually, model calibration refers to the consistency between the confidence score of the model output and the actual accuracy. As we mentioned several findings in RQ4 of Section 5.4.1 in the evaluation stage:
>
> - It is first evident that the confidence of GPT-4 increases through reflection steps, while LLMs show overconfidence at step-3 under different strategies.
>
> - It is observed that LLMs exhibit the highest level of confidence in settings where shortcuts (reminders) are easy to capture, consistent with findings in deep neural models.
>
> - Notably, despite the high confidence levels, *the overall confidence level of LLMs in settings with random cues is lower than that in settings with truth cues.* This is consistent with their performance accuracy.
>
> - The overall confidence of the LLMs at step-3 decreases under the guidance of different roles compared to the no-role guidance, which is similar to student performance and reflects their uncertainty.
>
> In addition, in the fine-tuning stage, the calibration ability of fine-tuned LLM has improved, and the gap between confidence scores and accuracy is narrowing as shown in Tables 4 and 5.
>
> ---
>
> **Comment7**: How was the verbal confidence level such as "very confident" converted to scores?
>
> **Answer**: Thanks for the detailed question! In Appendix A.2, we mentioned:
>
> > In the statistics of experiments, since there are few cases of non-numerical confidence levels and it is difficult to quantify, we compute on numerical confidence levels.
>
> If verbalized confidence is not expressed as a numerical value, confidence does not produce a progressive relationship like the numerical scores with the deepening of the reflection steps. Therefore, it is difficult for us to directly quantify it as a numerical value, which is not only unfair in numerical statistics but also fails in reflecting the confidence level of LLM directly. Considering the few cases (10-20%), we only compute on numerical confidence levels.
>
> ---

---

> > ### Author Response · Authors · 2024-11-20
> > **Response to Reviewer fznJ (4/4)**
> >
> > **Comment8**: Is there any evidence proving that this also holds for LLMs? This paper shows that LLMs are affected to different degrees by different types of guidance, but it does not directly build the link between "not really know" and "easily affected".
> >
> > **Answer**: Thanks for the valuable question! The paper is based on some findings in educational psychology, students who don’t really know are easily affected by teacher and peer guidance. In LLMs, "not really know" data can include two kinds of questions:
> >
> > - questions where LLM answers incorrectly or does not know the answer.
> >
> > - questions where LLM answers correctly but with low confidence, indicating it might change the answer and doesn’t truly know it.
> >
> > We extract the data in the two-step prompt setting (w/o RoSe setting). The data that LLM answers correctly in two steps is regarded as "know". and data that consistently results in incorrect answers over two steps or changes from correct to incorrect after reflection, is considered as "not really know".  The experimental results of the two groups of data under the RoSe strategy on EG-QA are as follows:
> >
> > The experimental results of RoSe strategy on "know" data:
> >
> > | Role | Reminder | Cue  | step-1 acc | step-1 conf | step-2 acc | step-2 conf | step-3 acc | step-3 conf | overall acc | overall conf |
> > | ---- | -------- | ---- | ---------- | ----------- | ---------- | ----------- | ---------- | ----------- | ----------- | ------------ |
> > | T    | ✓        | t    | 0.9921     | 0.8765      | 0.9942     | 0.9318      | 0.9964     | 0.9848      | 0.9942      | 0.9042       |
> > | T    | ✓        | r    | 0.9590     | 0.8797      | 0.9633     | 0.9321      | 0.9619     | 0.9727      | 0.9614      | 0.9059       |
> > | C    | ✓        | t    | 0.9814     | 0.8750      | 0.9828     | 0.9310      | 0.9864     | 0.9814      | 0.9835      | 0.9030       |
> > | C    | ✓        | r    | 0.9679     | 0.8826      | 0.9701     | 0.9357      | 0.9693     | 0.9772      | 0.9691      | 0.9092       |
> >
> > The experimental results of RoSe strategy on "not really know" data:
> >
> > | Role | Reminder | Cue  | step-1 acc | step-1 conf | step-2 acc | step-2 conf | step-3 acc | step-3 conf | overall acc | overall conf |
> > | ---- | -------- | ---- | ---------- | ----------- | ---------- | ----------- | ---------- | ----------- | ----------- | ------------ |
> > | T    | ✓        | t    | 0.4511     | 0.8358      | 0.4360     | 0.9064      | 0.4736     | 0.9584      | 0.4536      | 0.8711       |
> > | T    | ✓        | r    | 0.3106     | 0.8343      | 0.3030     | 0.9135      | 0.3030     | 0.9609      | 0.3055      | 0.8739       |
> > | C    | ✓        | t    | 0.3511     | 0.8373      | 0.3740     | 0.9007      | 0.3969     | 0.9477      | 0.3740      | 0.8690       |
> > | C    | ✓        | r    | 0.2519     | 0.8312      | 0.2290     | 0.8974      | 0.2137     | 0.9446      | 0.2315      | 0.8643       |
> >
> >
> >
> > Overall, GPT-4 shows a slight impact from different roles when processing the "know" data. However, when handling "not really know" data, the influence of different roles is more pronounced, resulting in a difference of over 15%. Besides, in the "not really know" data, the model's calibration ability is also worse, and the overall confidence level is lower compared to in the "know" data. Therefore, we can build the link between "not really know" and "easily affected" in LLMs: **LLMs are easily affected by role guidance when they "don't really know".** Meanwhile, we can also consider questions that are easily affected by role guidance and lead to changes in answers as "not really know" data.
> >
> > We will improve our paper based on all the constructive comments.

---

> > > ### Author Response · Authors · 2024-11-25
> > >
> > > Dear reviewer fznJ,
> > >
> > > As the open discussion period draws to a close in a few days, we want to check back to see whether you have any remaining concerns.  Thank reviewer fznJ again for engaging with our work thoughtfully and constructively. We have provided global responses for all reviewers to highlight several supplements to the paper. In addition, we also believe that we have sufficiently responded to your earlier queries on various aspects of this work, and we provide a short summary here for your convenience:
> > >
> > >
> > > - Specific, fine-grained evaluation method (RoSe strategy) and novel fine-tuning open-source LLMs (double-calibrated strategy).
> > > - The detailed introduction of RoSe strategy, calibrated fine-tuning, and well-calibrated data.
> > > - The reason for choosing EG-QA is to help with fine-tuning, without manual annotation, which is currently available [here](https://anonymous.4open.science/r/EG-QA-C2B2).
> > > - Clarify some confusion: logical consistency, model calibration ability, conversion of verbal confidence level into scores.
> > > - More experimental results confirm the link between "not really know" and "easily affected".
> > >
> > >
> > > Please let us know if/how we can address any remaining concerns, and we are grateful for any additional feedback and suggestions.
> > >
> > > Best,
> > >
> > > Authors

---

> > > > ### Comment · Reviewer_fznJ · 2024-11-25
> > > > **Thanks for your response**
> > > >
> > > > I would like to thank the authors for their responses. After carefully reading through your responses, the following concerns linger.
> > > >
> > > > - C1. Thanks for your response, but still, the issue exists.
> > > >
> > > > - C2. I appreciate your revision of the narration. It would be better if you could better illustrate the entire pipeline of the article, including the fine-tuning part.
> > > >
> > > > - C3. I don't think the reason is convincing enough for not choosing more widely-used datasets such as BBH or MATH. According to your shared dataset, it seems that all questions are in English. I wonder why you claim the dataset to be bilingual. In addition, I'm not sure whether you have noticed that a lot of Unicode white space characters are used in your dataset (marked as red in the anonymous GitHub). Would it cause some trouble (not errors, but incorrect word separations) in tokenization?
> > > >
> > > > - C4. I'm looking forward to seeing your code. It would be better if you could adapt your code to PyTorch and CUDA (I'm not sure whether PyTorch supports Ascend NPUs) for easier evaluation or adaptation of your method.
> > > >
> > > > - C5. Then why p and q are removed from the conditional terms?
> > > >
> > > > - C6. Please include calibration plots and ECE to justify your statement.
> > > >
> > > > - C7. Thanks for the explanation! It would be better if you could put it into the article as the original statement is somewhat confusing.
> > > >
> > > > - C8. Thanks for your explanation.

---

> ### Author Response · Authors · 2024-11-27
> **Response to Reviewer fznJ (1/4)**
>
> We sincerely appreciate your valuable time and feedback on our paper. Thanks for your thoughtful feedback and suggestions on our paper! We are committed to thoroughly addressing your concerns.
>
> **C1.** Thanks for your response, but still, the issue exists.
>
> **Answer**: Regarding the comment *"Although this article attends to more specific aspects of when and how LLM could fail, the demonstrated results are intuitive and may not deserve the discussion using a 10-page conference paper"*, we are sorry that our response did not satisfy. We would like to explain main contributions of our paper mainly from the significance and structure of the paper combined with reviewers' evaluation to verify that the paper is worth 10-page research, and we have much more than 10 pages of discussion:
>
> 1. Significance of the paper:
> - **The refreshing and interesting findings (Reviewers nbt9, JdDF, fznJ)**. We first propose the RoSe strategy to explore the ability of LLMs to "know what they know", and reveal the local information (strong reminder) that the model relies on most. The RoSe strategy can mitigate the learning on shortcut-reminder. Meanwhile, we discover the potential trust of LLMs on the authority role.
>
> - **Benefit a wide spectrum of the NLP community (Reviewers nbt9, fznJ)**. As reviewer agrees, the findings in the paper can promote more discovery of LLMs in the NLP community. Our proposed double-calibrated strategy can effectively fine-tune open-source LLMs. Without dedicated human annotation, it can effectively auto-obtain high-quality CoT processes by combining RoSe strategy with confidence calibration, which effectively helps improve the logical reasoning and calibration abilities of open-source LLMs. The effectiveness of the strategy is fully verified on various open-source LLMs and datasets. Moreover, we construct the EG-QA dataset, which will be helpful to the related community that needs English multiple-choice QA suite.
>
> - **Better align with real-world reasoning patterns, supporting more robust, real-world applications (Reviewer mKeV) & Convincing motivation (Reviewer nbt9).** Inspired by research in Educational Psychology that students who don't really know are easily affected by teacher and peer guidance, we treat LLM as a student to promote more research on evaluating and comparing LLMs with human behavior. We find similarities and differences between LLMs and human behavior, and these findings can further help improve the performance of LLMs.
>
>
> 2. Structure of paper: the structure of this paper is mainly divided into two aspects, evaluation of closed-source LLMs and fine-tuning of open-source LLMs.
>
> - **In the evaluation**, we propose the RoSe strategy from the perspective of educational psychology, which helps us explore several behaviors of LLMs. The findings on similarities and differences in LLMs compared to human behavior can be obtained by extensive experiments:
>
>   (1) Similar to human behavior, LLMs are easily affected by role guidance when they don't really know. In addition, LLMs tend to trust the role of authority more when guided by different roles.
>
>   (2) Unlike human behavior, LLMs exhibit over-reliance on strong reminder information due to the gradient training and training/SFT data distribution.
>
> - **In the fine-tuning**, we propose double-calibrated strategy to extract well-calibrated data and help fine-tune open-source LLMs:
>   (1) The first calibration is to obtain data that remains accurate during the self-reflection and the role-guidance process, which is unaffected by role-guidance and has a progressive and high-quality reasoning process through self-reflection.
>
>   (2) The second calibration is to obtain data that LLM "really knows" combined with confidence calibration. The model expresses confidence in its own answer in the reasoning process, without showing uncertainty affected by role guidance.
>
>   Double-calibrated strategy helps automatically obtain high-quality reasoning process, accurate answers, consistent confidence scores, and effectively helps optimize model parameters in the fine-tuning process to improve open-source LLMs' reasoning ability and maintain self-reflection ability.
>
> ---

---

> ### Author Response · Authors · 2024-11-27
> **Response to Reviewer fznJ (2/4)**
>
> **C2.** I appreciate your revision of the narration. It would be better if you could better illustrate the entire pipeline of the article, including the fine-tuning part.
>
> **Answer:**
> Thanks for your question! we explained the main structure of the paper in the previous answer. Two main (Rose, double-calibrated) strategies are proposed to help evaluate and fine-tune LLMs, respectively.
> Considering that the reviewer is mainly confused about fine-tuning, we would like to introduce the **process** and **goal** of fine-tuning in detail. The process of fine-tuning is to optimize the model parameters based on task-specific data. In the following fine-tuning objective optimization definition, we want to minimize the loss function, Let the gap between the predicted output of the model $M_{\theta}(q \oplus \wp)$ and the actual reasoning process $(r \oplus a \oplus c)$ be minimized.
>
> > $ \theta^* = \arg\min_{\theta} \mathcal{L}(M_{\theta}(q \oplus \wp), (r \oplus a \oplus c)).$
>
> In task-specific fine-tuning, it often requires high-quality human annotations, including detailed CoT processes ($r$) and answers ($a$). However, with the double-calibrated strategy, we can automate obtain $r \oplus a \oplus c$ from the strong closed-source LLM. Instead of simply focusing on the data that answers correctly, we focus on the data that still answers correctly during the self-reflection and the role guidance process, which contains high-quality reasoning. Meanwhile, coupled with confidence calibration ($c$), we can obtain the data that LLM "really knows".
>
> Furthermore, we mention in the main paper:
>
> > We propose thought-based and calibrated fine-tuning methods to align the Chain-of-Thought (CoT) process with corresponding confidence levels at each reflection step.
>
> Where the **thought-based method** is the fine-tuning model **alignment reasoning process $r$**, and **the calibrated fine-tuning** process is to fine-tune LLMs to **align answer $a$ and confidence $c$**, i.e., the model calibration capability. It also corresponds to the **double-calibrated strategy** we mentioned in the previous answer, respectively. Therefore, by acquiring such well-calibrated data, we can enable open-source LLMs to learn robust, high-quality CoT data during the fine-tuning process, while simultaneously improving the ability to self-reflect.
>
>
> ---

---

> ### Author Response · Authors · 2024-11-27
> **Response to Reviewer fznJ (3/4)**
>
> **C3.** I don't think the reason is convincing enough for not choosing more widely-used datasets such as BBH or MATH. According to your shared dataset, it seems that all questions are in English. I wonder why you claim the dataset to be bilingual. In addition, I'm not sure whether you have noticed that a lot of Unicode white space characters are used in your dataset (marked as red in the anonymous GitHub). Would it cause some trouble (not errors, but incorrect word separations) in tokenization?
>
> **Answer**: Thanks for your question! We have added additional instructions in the README file and would like to address both your questions and the importance of EG-QA.
>
> - Bilingual EG-QA: Since EG-QA contains English examinations **for Chinese students**, **the corresponding question stem** (clarifying the task, providing background information, setting requirements and constraints) **is in Chinese**, and some questions have stems that contain details about the exam, such as particular region and grade in which it took place. **We are not sure whether this violates the anonymity policy, so we removed the stem part**. In addition to the stem part, EG-QA also includes Chinese explanations for certain unfamiliar words. We can give some examples [here](https://anonymous.4open.science/r/EG-QA-C2B2) that do not violate the anonymity policy. When the EG-QA is officially released, we will disclose the whole dataset.
>
> - Data processing: We have normalized the white space characters to a standard space character (U+0020) before tokenization when loading data.
>
> - EG-QA: The core issue is the need for testing environments after fine-tuning LLMs:
>
>    1. Current QA datasets often cannot guarantee they weren't employed during pre-training or RLHF processes (**data contamination** issue commented by Reviewer nbt9). This undermines the model's ability to generalize and evaluate new or unseen data. If the model has memorized answers from the QA datasets used in pre-training, it could fail to properly reason through questions, simply recalling answers it has already encountered, rather than aligning through fine-tuning.
>
>    2. Existing benchmark **lacks clear ID and OOD test environment**, failing to properly evaluate how well a model generalizes to new, unseen data after fine-tuning and hindering the assessment of generalization after fine-tuning. Meanwhile, it will cause an **over-optimistic evaluation** problem, if only ID data is used for evaluation, the model may appear to generalize well, but in reality, it may only perform well on data similar to what it was trained on. This could lead to misleading conclusions about the model’s real-world performance. It is crucial to understand how well a model performs outside of its training environment by evaluating ID and OOD data.
>
>   The EG-QA dataset addresses these issues with newly collected data and contains diverse knowledge points, providing clear ID and OOD splits.
>
> ---
>
>
> **C4.** I'm looking forward to seeing your code. It would be better if you could adapt your code to PyTorch and CUDA (I'm not sure whether PyTorch supports Ascend NPUs) for easier evaluation or adaptation of your method.
>
> **Answer**: Thanks for your question! Since the paper has not yet been published and ICLR is an open platform, we found that there are already a few people looking at this repository, we are releasing the code for fine-tuning LLaMA3-8B [here](https://anonymous.4open.science/r/EG-QA-C2B2). We hope this will help! The code for fine-tuning models is all based on PyTorch. The experimental setup is detailed in section 5.2. Both Qwen and LLaMA3 are fine-tuned on A100-80G GPUs, and only Spark is fine-tuned on Ascend 910B 64G NPUs. The Ascend 910B also supports PyTorch, which can refer to https://github.com/Ascend/pytorch.
>
> Furthermore, the open-source community has already made fine-tuning widely and easily accessible, so fine-tuning itself is not the main contribution of the paper. One of key contributions of our work is in developing a double-calibrated strategy for extracting high-quality CoT processes, which are essential for improving the model's performance during fine-tuning. The fine-tuned data, models, and code will all be open-sourced after the paper is published (**mentioned in footnote 8**).
>
> ---
>
> **C5.** Then why p and q are removed from the conditional terms?
>
> **Answer**: Thanks for your detailed question! We want to emphasize logical consistency, i.e., the model response is updated after each reasoning step based on the current reasoning result. The internal answer and confidence scores are consistent with its CoT process, which is also a reflection of the model's calibration ability.
>
> However, considering the confusion caused by the logical consistency $P(a,c|r)$, we have modified it in the paper (highlighted in green).
>
> ---

---

> ### Author Response · Authors · 2024-11-27
> **Response to Reviewer fznJ (4/4)**
>
> **C6.** Please include calibration plots and ECE to justify your statement.
>
> **Answer**: Thanks for your constructive suggestion! The calibration plots of GPT-4 turbo and ECE scores of LLaMA3-8B and Spark-13B are supplemented in **Appendix B.8** (highlighted in green).
>
> Considering your constructive suggestion, to better compare the calibration abilities of LLMs in the evaluation and after fine-tuning, we employ calibration plots and ECE scores to perform evaluations on GPT-4 turbo and fine-tuned LLaMA3-8B, Spark-13B.
> First, the calibration plots of GPT-4 are shown in Figure 9. Overall, **the confidence levels of the LLM are high**, typically above 60%, with most values falling to the lower right side of the perfectly calibrated line. As shown in the top row, under various RoSe strategies, GPT-4 demonstrates good calibration performance at step-1, but its performance declines at step-3 under role guidance. Meanwhile, the calibration performance of the LLM at step-3 is worse under random answer guidance, which is consistent with the findings of RQ1 in Section 5.4.1.
>
> Furthermore, as shown in the bottom row of Figure 9, compared with the model calibration performance under the RoSe strategy, **LLM exhibits poorer ability without role guidance, which aligns with the findings in RQ3 of Section 5.4.1.**
> LLMs tend to capture shortcuts by relying solely on strong reminder in prompts to quickly find the answer, role guidance can reduce the over-reliance of LLMs on reminders to a certain extent.
>
>
> Then, we employ Expected Calibration Error (ECE) by comparing the confidence scores with the actual accuracy of the predictions for evaluation. As shown in Figures 10 and 11, **the fine-tuned LLaMA3-8B and Spark-13B generally have better calibration abilities than base LLMs**. The calibration abilities of LLMs at step-2 are worse than the other two steps, indicating that LLMs modify their correct answers at step-1 and improve their confidence scores during the self-reflection process, leading to their higher ECE scores. This also demonstrates the significance of evaluating whether LLMs know what they know in Section 5.4.1, where LLMs fail to adhere to their correct answer, exhibiting uncertainty but accompanied by rising self-confidence.
>
> Moreover, under the guidance of classmate and random cue information, the fine-tuned LLMs exhibit poor calibration ability compared with base LLMs at step-1 and step-2, there appears to be a discrepancy between the models' high confidence levels and their actual accuracy. After two steps of reflection and adjustment, there is a notable enhancement in the models' performance at step-3, which suggests the alignment of increased accuracy and confidence levels.
>
>
> ---
>
> **C7.** Thanks for the explanation! It would be better if you could put it into the article as the original statement is somewhat confusing.
>
> **Answer**: Thanks for your insightful question! we have supplemented the explanation in **Appendix A.2** (highlighted in green).
>
> ---
>
> Thank reviewer fznJ again for engaging with our work thoughtfully and constructively, and we are grateful for any additional feedback and suggestions.

---

> > ### Comment · Reviewer_fznJ · 2024-11-27
> >
> > Thanks for your explanation. I've raised my score to positive.

---

> > > ### Author Response · Authors · 2024-11-27
> > >
> > > We greatly appreciate your positive comments and acknowledgment of our paper! We will carefully incorporate these clarifications and further improve the quality of our paper.

---

### Official Review · Reviewer_mKeV · 2024-10-25

**Soundness:** 2
**Presentation:** 3
**Contribution:** 2
**Rating:** 6
**Confidence:** 3

**Summary:**

This paper focuses on testing and boosting the model’s self-knowledge. its ability to tell the difference between what it truly understands and what it’s guessing from training data, rather than just following prompts or role guidance.
They’re using different authoritative roles, like teacher or judge, to see how the model responds in each role, but the goal isn’t to pick one set role for guiding it permanently.

So, the aim is to check if the model falls for misleading cues, especially when it doesn’t actually know something. By introducing these authoritative roles, the researchers can see if the model just goes along with what it’s told.
This lets them understand how the model behaves in different scenarios and figure out the kinds of guidance that might encourage more independent thinking.

**Strengths:**

- The authors implement role guidance by assigning roles, like "teacher" or "judge," to help the model think in ways that better align with real-world reasoning patterns.
- Adding a self-reflection step enables the model to review its responses, which enhances accuracy and reliability while exploring its self-knowledge.
- The paper’s double-calibration strategy combines role guidance with self-reflection, adjusting prompts and roles in iterative steps to reduce susceptibility to misleading information and improve answer stability.
- This approach also offers finer control during fine-tuning, helping the model handle uncertain information without relying solely on intuition or single-step decisions.
- The authors emphasize model self-knowledge, designing experiments to observe its confidence levels under different conditions. This focus helps develop models that are both accurate and capable of self-assessment, supporting more robust, real-world applications.

**Weaknesses:**

- This work mostly used random answers to mislead the model, but they didn’t explain in detail how these answers were generated to ensure they’re diverse and realistic. If the random answers are too simple or repetitive, they may not truly test how well the model can handle more challenging misleading cues.
- The misleading information in the tests was mostly straightforward or basic incorrect answers. But in real-world scenarios, misleading information is often subtler or harder to detect. This setup might not fully capture the challenges the model would face in real-life situations.
- The model’s self-knowledge is mainly judged by its confidence levels and accuracy. These indicators alone might not be enough to fully capture how well the model truly understands its answers.
- Roles like “judge” often require objectivity and caution, which might make the model more conservative in its responses. This cautious approach could limit the model’s effectiveness, especially in tasks that require flexible reasoning or hypothesis testing.




It may be helpful to reference the following papers and incorporate a discussion:
- Xie, Jian, et al. "Adaptive Chameleon or Stubborn Sloth: Revealing the Behavior of Large Language Models in Knowledge Conflicts." The Twelfth International Conference on Learning Representations.

It looks like that paper (Jian, et al.) also digs into situations where LLMs can be misled. Could the authors add some extra insights by comparing those findings with their own experiments here?
The experimental results in this paper feel a bit limited when it comes to offering new perspectives.

- Chan, Chi-Min, et al. "ChatEval: Towards Better LLM-based Evaluators through Multi-Agent Debate." The Twelfth International Conference on Learning Representations.

**Questions:**

- Are the randomly generated answers diverse and realistic enough to really test the model’s ability to handle complex misleading situations?
- Can the misleading info in the experiment truly reflect the subtle or hidden misdirections found in real-world scenarios to fully test the model's response?
- By relying just on confidence levels and accuracy to measure the model's self-awareness, are we capturing the full depth of its understanding?
- And could roles like a "judge" make the model more conservative, possibly affecting its performance on tasks that need flexible reasoning or hypothesis testing?

---

> ### Author Response · Authors · 2024-11-20
> **Response to Reviewer mKeV (1/3)**
>
> We thank the reviewer for the thoughtful and detailed comments. We are pleased that the reviewer considers our research that better align with real-world reasoning patterns. We appreciate the opportunity to address the concerns here.
>
> ---
>
> **Comment1**: This work mostly used random answers to mislead the model, but they didn’t explain in detail how these answers were generated to ensure they’re diverse and realistic.
>
> **Answer**: Thank you for the insightful comment. Our primary test scenario aligns with the findings in educational scenarios that "students who don't really know are easily affected by teacher and peer guidance". We collect EG-QA and employ JEC-QA as test suites comprising multiple-choice examination questions, tailored for middle school students and law candidates, respectively.
>
> We make evaluations on multiple-choice QA task, answers are usually identified by a letter (e.g., A, B, C,  D). Therefore, both truth and random answers are letters, which ensures the experimental results' stability, consistency, and reproducibility. In the evaluation of random answers, we utilize a fixed seed to generate random letters for each question, the probability that the generated answer differs from the correct answer is about 75%.
>
> ---
>
> **Comment2**: The misleading information in the tests was mostly straightforward or basic incorrect answers. But in real-world scenarios, misleading information is often subtler or harder to detect. This setup might not fully capture the challenges the model would face in real-life situations.
>
> **Answer**: Thanks for your insightful suggestion! In this paper, we treat the LLM as the student, so we evaluate the performance of LLMs in both English and domain-specific law examinations.
>
> Referencing to prompt information can be subtle in real-world scenarios as the reviewer suggests. Meanwhile, considering that the answers provided by the teacher or the classmate are not independent of the question itself, the LLM will not believe the information provided by the role guidance if it is not related to the question itself. We substitute the the letter options in prompts into their corresponding textual descriptions (each letter corresponds to a specific text content or answer description). This transformation makes the prompt information more complex and requires deeper understanding and processing by LLMs.
>
> The experimental results on EG-QA are as follows, where $t_c$ and $r_c$ denote the textual descriptions of truth and random answer:
>
> | Role | Reminder | Cue   | step-1 acc | Δ       | step-1 conf | step-2 acc | Δ       | step-2 conf | step-3 acc | Δ       | step-3 conf | overall acc | overall conf |
> | ---- | -------- | ----- | ---------- | ------- | ----------- | ---------- | ------- | ----------- | ---------- | ------- | ----------- | ----------- | ------------ |
> | w/o  | ✗        | ✗     | 0.9108     | -       | **0.8889**  | 0.9159     | -       | **0.9676**  | -          | -       | -           | 0.9134      | 0.9283       |
> | T    | ✓        | t     | **0.9431** | +0.0323 | 0.8726      | **0.9450** | +0.0291 | 0.9295      | **0.9494** | +0.0334 | 0.9825      | **0.9458**  | 0.9282       |
> | T    | ✓        | r     | _0.9070_   | -0.0038 | 0.8752      | 0.9108     | -0.0051 | 0.9302      | 0.9101     | -0.0058 | 0.9716      | 0.9093      | 0.9257       |
> | C    | ✓        | t     | 0.9322     | +0.0214 | 0.8717      | 0.9335     | +0.0176 | 0.9287      | 0.9373     | +0.0213 | 0.9785      | 0.9343      | 0.9263       |
> | C    | ✓        | r     | 0.9085     | -0.0023 | 0.8781      | _0.9092_   | -0.0067 | 0.9325      | _0.9067_   | -0.0092 | 0.9741      | _0.9081_    | 0.9282       |
> | T    | ✓        | $t_c$ | **0.9390** | +0.0282 | 0.8796      | **0.9433** | +0.0274 | 0.9328      | **0.9457** | +0.0298 | 0.9715      | **0.9427**  | 0.9062       |
> | T    | ✓        | $r_c$ | _0.8700_   | -0.0408 | 0.8810      | _0.8688_   | -0.0471 | 0.9312      | _0.8639_   | -0.0520 | 0.9623      | _0.8676_    | 0.9061       |
> | C    | ✓        | $t_c$ | 0.9194     | +0.0086 | 0.8852      | 0.9219     | +0.0060 | 0.9365      | 0.9225     | +0.0066 | 0.9698      | 0.9212      | 0.9109       |
> | C    | ✓        | $r_c$ | 0.8741     | -0.0367 | 0.8885      | 0.8796     | -0.0363 | 0.9372      | 0.8778     | -0.0381 | 0.9679      | 0.8772      | 0.9128       |
>
> Under the influence of subtle cue information, the overall performance of GPT-4 is lower than that under letter cue information, and the overall conclusions of the experimental results are consistent with the findings in Section 5.4.1. Compared to letter options, LLM is less sensitive to text cue information, and it is difficult to associate it with the option content in the original question. Since it cannot distinguish between relevant and misleading information, it causes LLM to become distracted and unable to reason and answer questions effectively. We analyze the experimental results in Appendix B.7 of the updated version.

---

> ### Author Response · Authors · 2024-11-20
> **Response to Reviewer mKeV (2/3)**
>
> **Comment3**: The model’s self-knowledge is mainly judged by its confidence levels and accuracy. These indicators alone might not be enough to fully capture how well the model truly understands its answers.
>
> **Answer**: Thanks for the valuable suggestion! In this paper, we make evaluations and propose double-calibrated strategy to combine accuracy and confidence scores. The well-calibrated data could help extract high-quality reasoning data to improve the reasoning ability of open-source LLMs. Considering the suggestions of the reviewer, we further evaluate the internal consistency of the LLM in the self-reflection process, we make a detailed analysis in Appendix B.6 of updated version.
>
> First, we employ GPT-4 ($Con_{GPT}$) and human annotation ($Con_{human}$) to evaluate the internal consistency between the reasoning steps of the model on the challenging samples. We consider the samples where the LLM makes errors during the two-step reflection process to be challenging. In EG-QA, approximately 8% of the data consists of these challenging samples, which are not really known to the LLM.
>
> We utilize the prompt in Table 14 (in the updated version) and human annotation to evaluate the reasoning consistency of LLM in three steps (In step-1, we prompt LLMs output answers. In step-2, we prompt LLMs to self-reflect on the answer of the previous step and further answer the question. In step-3, we employ different role guidance to evaluate the performance of LLMs.).
>
> Specifically, the consistency between step-1 and 2 shows little difference between GPT-4 and human annotations. However, on the consistency between step-2 and 3, the human annotations demonstrate higher consistency. Although the logical expression from step-2 to step-3 is consistent, GPT-4 annotations tend to focus more on semantic consistency, often overlooking the progression of logical expression. Overall, the logical reasoning in the self-reflection process across the three steps is consistent for LLMs.
>
> Then, in $Guidance$ of experimental results, we manually annotate whether the responses of these challenging samples in step-3 follow the role guidance information. Consistent with findings in **RQ3 of Section 5.4.1**, LLMs tend to trust the role of authority more and more easily affected by authority-teacher.
>
> | Role | Reminder | Cue  | step-1&2 $Con_{GPT}$ | step-1&2 $Con_{human}$ | step-2&3 $Con_{GPT}$ | step-2&3 $Con_{human}$ | step-3 $Guidance$ |
> | ---- | -------- | ---- | -------------------------- | ---------------------------- | -------------------------- | ---------------------------- | --------------- |
> | T    | ✓        | t    | 0.9548                     | 0.9473                       | 0.7669                     | 0.9248                       | 0.4210          |
> | T    | ✓        | r    | 0.9545                     | 0.9772                       | 0.7954                     | **0.9924**                   | **0.4318**      |
> | C    | ✓        | t    | **0.9923**                 | **0.9923**                   | 0.8778                     | 0.9618                       | 0.3816          |
> | C    | ✓        | r    | 0.9236                     | 0.9312                       | **0.8854**                 | 0.9923                       | 0.2213          |
>
> ---
>
>
> **Comment4**: Roles like “judge” often require objectivity and caution, which might make the model more conservative in its responses. This cautious approach could limit the model’s effectiveness, especially in tasks that require flexible reasoning or hypothesis testing.
>
> **Answer**: Thanks for the constructive question! As illustrated in line 277-281 (revision line 274-279), the test scenario is mainly on multiple-choice questions in the context of legal professional examination. We mainly evaluate on Knowledge-Driven questions (KD-questions), which mainly focus on the **fixed knowledge** corresponding to the law articles, such as civil law, commercial law, criminal law, etc. When dealing with case analysis questions, LLMs might respond more conservatively. However, for questions relying on established knowledge, they don’t need to be conservative and should focus on selecting the appropriate answer.
>
> ---

---

> ### Author Response · Authors · 2024-11-20
> **Response to Reviewer mKeV (3/3)**
>
> **Comment5**: It may be helpful to reference the following papers and incorporate a discussion.
>
> > [1] Xie, Jian, et al. "Adaptive Chameleon or Stubborn Sloth: Revealing the Behavior of Large Language Models in Knowledge Conflicts." ICLR2024.
>
> > [2] Chan, Chi-Min, et al. "ChatEval: Towards Better LLM-based Evaluators through Multi-Agent Debate." ICLR2024.
>
>
> **Answer**: Thanks for your critical suggestion! Actually, [2] mainly employs multi-agent debate for text evaluation tasks. Existing text evaluation tasks mainly rely on manual annotation, the paper employs multiple LLMs to evaluate the quality of text instead of evaluating the ability of LLM. The direction of our research is different. Then, refer to the paper [1], we will discuss it in Section 2 in the updated version. Specifically, we are different in four aspects: motivation, task, evaluation method and evaluation object as follows:
>
>
>
> - Motivation: [1] mainly studies the conflict between external knowledge and parameterized memory of LLMs in RAG scenarios. However, we mainly evaluate whether LLMs really know what they know and what they do not know to better ensure trustworthy in real-world scenarios.
>
> - Task: [1] makes evaluations on entity substitution QA (POPQA) and reasoning for providing True or False answers (STRATEGYQA). We mainly employ EG-QA and JEC-QA, which both focus on multiple-choice questions in educational scenarios, aligning with the motivation of the paper.
>
> - Method: [1] elicits the parametric memory of LLMs and generates coherent counterfactual constructs by substituting entities, further evaluating the model's acceptance of parametric and counterfactual knowledge.
>
>     Our approach primarily introduces Role-guided and Self-reflection (RoSe) strategy in prompts to assess the model's ability to self-improve within an individual feedback and its "know what they know" capability. Furthermore, we employ a double-calibrated strategy to extract high-quality reasoning processes, which helps fine-tune open-source LLMs, enhancing their reasoning abilities and reducing their focus on strong reminder information.
>
> - Object: [1] focuses on parametric memory and filters out the inconsistent data of model answers through entailment checking and answer consistency, which is called unqualified examples. The data [1] chooses not to learn is the data that we focus on "not really know", that is, the model answer is changed from correct to incorrect after reflection or confused by role-guided information.
>
>
> We will improve our work based on all the constructive comments.

---

> > ### Author Response · Authors · 2024-11-25
> >
> > Dear reviewer mKeV,
> >
> > As the open discussion period draws to a close in a few days, we want to check back to see whether you have any remaining concerns. Thank reviewer mKeV again for engaging with our work thoughtfully and constructively. We have provided global responses for all reviewers to highlight several supplements to the paper. In addition, we also believe that we have sufficiently responded to your earlier queries on various aspects of this work, and we provide a short summary here for your convenience:
> >
> > - The answer settings in multiple-choice QA tasks.
> > - More experiments with subtle cue information (Appendix B.7).
> > - More evaluation metrics by GPT-4 and human annotation are employed to assess the internal consistency (Appendix B.6).
> > - The KD-questions settings on JEC-QA.
> > - The reference and discussion of the works recommended by the reviewer.
> >
> > Please let us know if/how we can address any remaining concerns, and we are grateful for any additional feedback and suggestions.
> >
> > Best,
> >
> > Authors

---

> > > ### Comment · Reviewer_mKeV · 2024-11-26
> > >
> > > Thanks for the author's response. However, I still have some concerns:
> > >
> > > Whether the random answer generation can fully simulate misleading situations in the real world is still worth discussing.
> > >
> > > In real-world applications, LLMs might face guidance from more diverse roles. It is still necessary to test and discuss whether such role diversity would affect the model's response to misleading information.
> > >
> > > Additionally, misleading information in the real world might also involve multi-layered factors such as context and implication.

---

> ### Author Response · Authors · 2024-11-27
> **Response to Reviewer mKeV**
>
> We sincerely appreciate your valuable time and feedback on our paper. Thanks for the questions and suggestions! We are committed to thoroughly addressing your concerns.
>
> **C1.** Whether the random answer generation can fully simulate misleading situations in the real world is still worth discussing.
> &
> **C3.** Additionally, misleading information in the real world might also involve multi-layered factors such as context and implication.
>
> **Answer**:  Thanks for your questions! We fully agree that misleading is an important topic, and there is a specialized research field dedicated to solving this problem [1]. However, we would like to clarify that the focus of the paper is not misleading LLMs, nor does the word "misleading information" appear throughout the paper. **In this paper, our focus is on evaluating whether LLM really knows, beyond merely assessing their susceptibility to misleading information.**
>
> **[1]** Chen C, Shu K. Combating misinformation in the age of llms: Opportunities and challenges. AI Magazine, 2024, 45(3): 354-368.
>
> In this paper, inspired by **Educational Psychology**, we treat LLM as a student to promote more research on evaluating and comparing LLMs with human behavior. The interesting findings on similarities and differences in LLMs compared to human behavior can be obtained by extensive evaluation experiments in **multiple-choice QA test suite**:
>
> - Similar to human behavior, LLMs are easily affected by role guidance when they don't really know. In addition, LLMs tend to trust the role of authority more when guided by different roles.
> - Unlike human behavior, LLMs exhibit over-reliance on strong reminder information due to the gradient training and training/SFT data distribution.
>
> We are glad you appreciate the paper better aligns with real-world reasoning patterns, supporting more robust, real-world applications (Reviewer mKeV). Moreover, other reviewers appreciate the convincing motivation (Reviewer nbt9), the refreshing and interesting findings (Reviewers nbt9, JdDF, fznJ), and the benefit of a wide spectrum of the NLP community (Reviewers nbt9, fznJ).
>
> ---
>
> **C2.** In real-world applications, LLMs might face guidance from more diverse roles. It is still necessary to test and discuss whether such role diversity would affect the model's response to misleading information.
>
>
> **Answer:** Thanks for your suggestions! Indeed, to gain deeper insights, we expanded our evaluation to include role guidance from additional roles such as **Judge** and **Lawyer**. We incorporated nine different settings using the open-source legal multiple-choice QA dataset (JEC-QA) to thoroughly evaluate the influence of these roles. The experimental results can be found in Table 3, with detailed findings discussed in Section 5.4.1.
>
> ---
>
> Thank reviewer mKeV again for engaging with our work thoughtfully and constructively, and we are grateful for any additional feedback and suggestions.

---

> > ### Comment · Reviewer_mKeV · 2024-11-27
> >
> > Thank you for your response and clarification. With the updated information, I have revised my score.

---

> > > ### Author Response · Authors · 2024-11-27
> > >
> > > We greatly appreciate your positive comments and acknowledgment of our paper! We will carefully incorporate these clarifications and further improve the quality of our paper.

---

### Official Review · Reviewer_JdDF · 2024-11-03

**Soundness:** 3
**Presentation:** 2
**Contribution:** 3
**Rating:** 6
**Confidence:** 4

**Summary:**

This paper proposes RoSe, a strategy that uses role guidance and self-reflection in prompts to evaluate whether LLMs know what it knows. They use a double calibrated strategy to find well-calibrated data to be used for fine-tuning LLMs. They study four research questions and found some interesting observations. For example, LLMs are highly sensitive to strong reminder information in prompts, such as "the answer is". In addition, role guidance can reduce the issue of overconfidence of LLMs.

**Strengths:**

- The idea of using roles like "teacher", "student" and "classmate" is interesting.
- The authors provide a lot of details for reproducing the experiments, such as prompts for each step and experiment results under different settings.
- The findings of the paper are quite interesting but not surprising. For example, LLMs may be confused by wrong guidance, tend to capture information from shortcuts, and their overconfidence can be mitigated by role guidance.

**Weaknesses:**

- The writing of the paper is a bit unclear. The paper did not mention explicitly in the main method section about what are "role-guided", "self-reflection", and they only use a figure in the introduction to show what the prompt looks like.
- The author did not explain what is "conf" in Table 2 and Table 3. This is not a typical metric and the authors should explain why it is important.

**Questions:**

- Could you explicitly explain the "Role", "Rem", "Cue", "conf", "com" appearing in the experiment result table?

**Details Of Ethics Concerns:**

No ethics concerns.

---

> ### Author Response · Authors · 2024-11-20
> **Response to Reviewer  JdDF**
>
> We are grateful for your thoughtful feedback on our paper and happy to learn that you find our research detailed and the insights we provide interesting! We will address your concerns as follows.
>
> ---
>
> **Comment1**: The paper did not mention explicitly in the main method section about what are "role-guided", "self-reflection".
>
> **Answer**: We are sorry for any confusion on the terms "role-guided" and "self-reflection." Specifically, in line 76-81, motivated by some research in Educational Psychology, when students are not confident in their own performance (not really know), teacher and peer guidance may lead students to give up independent and in-depth thinking. In this paper, we treat the LLM as a student, incorporate **role guidance** with self-reflection in the prompt, explore what information the LLM depends on in several prompt settings, and *whether role guidance really shakes up the performance of LLM*.
>
> Then, in line 194-196, we introduce the **self-reflection** strategy, which involves three steps that prompt the LLM to reflect deeply on its response while *verifying whether it consists in the correct response*.
>
> We prompt LLMs self-reflective in the second and third steps and incorporate role guidance in the third step. The relevant prompt is as follows:
>
> > step 1: Please read the questions and options carefully and give the most appropriate answers and confidence;
>
> > step 2: Please read the questions and options carefully, continue to think, **reflect on the answer of step 1**, give the most appropriate answer and confidence;
>
> > step 3: My teacher thinks the answer is \{Truth\}. Please read the questions and options carefully, continue to think, **reflect on the answer of step 2**, and give the most appropriate answer and confidence.
>
> More prompt settings in step-3 can be found in Table 7.
>
> ---
>
> **Comment2**: What is "conf" in Table 2 and Table 3. This is not a typical metric and the authors should explain why it is important.
>
> **Answer**: Thanks for your pertinent question! "conf" refers to the verbalized **conf**idence in the paper. Specifically, since the log-probabilities of LLMs represent uncertainty over tokens (ways of expressing a claim) and not epistemic uncertainty over claims themselves, verbalized confidence is proposed by openAI [1] to elicit confidence in LLM and estimate LLM's confidence in their responses. We employ verbalized confidence to assess the model’s self-knowledge (really know) which is discussed in line 99, 147-152, and 159-160.
>
> We employ the prompt to ask LLM to output the confidence score at each step such as "give the most appropriate answer and confidence". It can be represented in percentage terms or using explicit descriptors like "high". In Appendix A.2, we mentioned:
>
> > In the statistics of experiments, since there are few cases of non-numerical confidence levels and it is difficult to quantify, we compute on numerical confidence levels.
>
> By integrating accuracy with confidence scores, we can better assess the model’s self-awareness of its knowledge and enhance the calibration ability of LLMs.
>
> [1] Lin S, Hilton J, Evans O. Teaching Models to Express Their Uncertainty in Words. Transactions on Machine Learning Research, 2022.
>
> ---
>
> **Comment3**: Could you explicitly explain the "Role", "Rem", "Cue", "conf", "com" appearing in the experiment result table?
>
> **Answer**: Thanks for your valuable suggestion! The explanations for "conf" and "acc" are supplemented on the captions of Tables, which can be found in line 325-329 of the updated version. In the answer to Comment2, we explain the concept of "conf" (verbalized confidence). Here we detailedly answer the reviewer's question to explain the meaning of "Role", "Rem", "Cue", "com":
>
> "Role", "Rem", "Cue" are important elements in role guidance prompts. Specifically, except that no role, **"Role"** could be “teacher” or “classmate” in educational scenarios, and also could be "Judge" or "lawyer" in legal scenarios. **"Rem"** is the abbreviation for "**Rem**inder". In RoSe strategy, the strong reminder is "answer is". **"Cue"** information represents the answer corresponding to the question, which could be the ground-truth or random answer (could be found in line 84-86).
>
>
> **$com$** is a new metric defined as the comprehensive completion degree in line 467-469 (revision line 470-473). Since open-source base LLMs usually cannot give a definite answer in step-1 and step-2 during experiments, exhibiting task avoidance [1]. To make fair comparisons, considering accuracy x and completion degree $C$ ($C$ refers to the proportion of LLM that gives the exact answer), we adopt the variant of F1-scores as evaluation on $com$: $2 \times \frac{A \times C}{A+C}$.
>
> [1] Zhou L, Schellaert W, Martínez-Plumed F, et al. Larger and more instructable language models become less reliable. Nature, 2024.
>
> We will improve our work based on all the constructive comments.

---

> ### Author Response · Authors · 2024-11-25
>
> Dear reviewer JdDF,
>
> As the open discussion period draws to a close in a few days, we want to check back to see whether you have any remaining concerns.  Thank reviewer JdDF again for engaging with our work thoughtfully and constructively. We have provided global responses for all reviewers to highlight several supplements to the paper. In addition, we also believe that we have sufficiently responded to your earlier queries on various aspects of this work, and we provide a short summary here for your convenience:
>
>
> 1. The detailed introduction of role-guided and self-reflection strategy.
> 2. The detailed introduction of metrics on verbalized confidence.
> 3. The explanations for "conf", "acc", "role", "rem", and "cue" have been supplemented on the captions of Tables.
>
> Please let us know if/how we can address any remaining concerns, and we are grateful for any additional feedback and suggestions.
>
> Best,
>
> Authors

---

> > ### Comment · Reviewer_JdDF · 2024-11-28
> >
> > Thanks for your explanation. It has addressed my concerns.

---

> > > ### Author Response · Authors · 2024-11-29
> > >
> > > We greatly appreciate your positive comments and acknowledgment of our paper! We have carefully incorporated these clarifications and further improve the quality of our paper.

---

### Official Review · Reviewer_nbt9 · 2024-11-03

**Soundness:** 3
**Presentation:** 3
**Contribution:** 3
**Rating:** 6
**Confidence:** 4

**Summary:**

This paper studies and evaluates whether large language models (LLMs) are confident in their acquired knowledge. It claims that LLMs fine-tuned with RLHF could potentially over-rely on aligned preferences, instead of truly gaining the knowledge, and that if LLMs are more confident in the knowledge. To qualitatively assess whether LLMs have a sense of whether it has any knowledge, a Role-Guided and Self-Reflection (RoSe) method is proposed. Specifically, it combines prompting and self-reflection to examine the sensitivity of LLMs to parametric knowledge and contextual knowledge. In the paper, several findings are elaborated. For example, empirical results reveal the LLMs are sensitive to the prompt. By assuming roles, LLMs are prone to be less dependent on the contextual knowledge. Based on the findings, the authors further propose a calibration-based method to extract high-quality SFT data. Fine-tuning on the SFT data improves the overall confidence when LLMs generate outputs.

**Strengths:**

- The motivation is convincing. Previous studies have revealed that deep learning models suffer from confidence calibration. To assess the confidence level of LLMs is an important topic and would benefit a wide spectrum of the NLP community.
- The experimental results are quite interesting and the findings are refreshing.

**Weaknesses:**

- Some details seem not clear to me. For example, what is exactly the _verbalized confidence_?
- It seems that the fine-tune portion of the experiments are all conducted on the EG-QA dataset, which is proposed in this submission as well. Whether the dataset suffer from data contamination needs serious examination.
- The proposed method mainly considered three factors to examine the confidence of LLM outputs (role, cue, etc.) There could be various other factors that have impact on the confidence (pre-trainining data, SFT data, preference data). Massive amount of studies on these factors might be needed to compose a "comprehensive" study.
- I think Section 4.2 could use some improvement. After reading it, it is still unclear to me how to conduce the so-called "double-calibration". I suggest the authors use some examples or diagrams to further illustrate.

**Questions:**

- What is exactly the _verbalized confidence_?
- How is the _double calibration_ achieved?
- How are the new datasets curated? How to make sure they are of high quality?
- What is step-3 in the experiment section?

---

> ### Author Response · Authors · 2024-11-20
> **Response to Reviewer nbt9 (1/2)**
>
> We thank the reviewer for the thoughtful and detailed comments. We are pleased that the reviewer considers our convincing motivation and refreshing findings! We appreciate the opportunity to address the concerns here.
>
> ---
>
> **Comment1&5**: what is exactly the verbalized confidence?
>
> **Answer**: We are sorry that the concept of “verbalized confidence” confused you. Based on the concept of Verbalized Calibration [1] first introduced by OpenAI, they find: since the log-probabilities of models like GPT-3 represent uncertainty over tokens (ways of expressing a claim) and not epistemic uncertainty over claims themselves, GPT-3 can learn to express calibrated uncertainty using words ("verbalized probability"), i.e. express uncertainty in the language ("61%" or "medium confidence").
>
> For verbalized confidence, we note that humans are able to verbalize their uncertainty, e.g., giving insight as to whether our answers and reasonings are correct or not. It is essential for LLMs to have the ability to know what they know rather than solely relying on data statistics. In Verbalized Confidence of Section 2, we introduce some related work in detail. Specifically, recent works on verbalized confidence aim that a trustworthy real-world prediction system should produce well-calibrated confidence scores. In our paper, we employ verbalized confidence to solve two problems:
>
> - Perform a more comprehensive evaluation, verbalized confidence helps us recognize the extent to which LLM knows the problem on its own, as we mentioned in line 45-48.
>
> - Verbalized confidence helps obtain high-quality data to fine-tune open-source LLMs, as we mentioned in line 237-238 (revision line 233-234).
>
> [1] Lin S, Hilton J, Evans O. Teaching Models to Express Their Uncertainty in Words. Transactions on Machine Learning Research, 2022.
>
> ---
>
> **Comment2&7**: Whether the EG-QA dataset suffer from data contamination needs serious examination.
>
>
> **Answer**: Thanks for the insightful question! As we mentioned in Section 5.1, the EG-QA dataset primarily consists of standardized examination questions collected from [website](http://www.zxxk.com/), which is an authoritative educational resources website, designed for Chinese teachers and students to provide teaching resources, learning resources, examination papers, teaching courseware and other text materials. The data in EG-QA include questions, standard answers, and knowledge points, requiring no additional manual annotation. The data originates from after **December 2023**, ensuring high quality and no data contamination issues. EG-QA is currently available [here](https://anonymous.4open.science/r/EG-QA-C2B2).
>
> ---
>
> **Comment3**: There could be various other factors that have impact on the confidence (pre-trainining data, SFT data, preference data). Massive amount of studies on these factors might be needed to compose a "comprehensive" study.
>
> **Answer**: Thanks for your valuable question! Evaluating LLMs is inherently complex, with varying motivations, goals, and methods. Therefore, we evaluate LLMs from the perspective of **comparative study** through role guidance and self-reflection strategy. In addition to the factors at the data level mentioned by the reviewer, LLMs are also affected by model gradient training. In this paper, we obtain some findings both at the data level and model training level:
>
> - At the data level, LLMs are alignment with human preferences for safety and trust may make LLMs more attentive to human concerns in the SFT stage. Despite extensive efforts to reduce bias in SFT data (preference data), our work shows that LLMs are still prone to trust authoritative roles (in RQ3 of Section 5.4.1). Data-level bias is still pervasive, it is significant to mitigate the hidden biases in LLMs. As shown in Tables 4, 5, 6, the double-calibrated strategy proposed to fine-tune open-source LLMs can mitigate the bias on authoritative roles caused by SFT data.
>
> - At the training level, the gradient training can lead models to find shortcuts. Since given strongly-correlated and fast-to-learn features in training data, gradient descent is biased toward learning them first [1]. As we illustrate in RQ2 of Section 5.4.1: LLMs tend to capture shortcuts by relying solely on strong reminder "answer is" in prompts to quickly find the answer rather than understanding genuine relationships between prompt and truth during training.
>
> [1] Pezeshki M, Kaba O, Bengio Y, et al. Gradient starvation: A learning proclivity in neural networks. Advances in Neural Information Processing Systems, 2021.

---

> > ### Author Response · Authors · 2024-11-20
> > **Response to Reviewer nbt9 (2/2)**
> >
> > **Comment4&6**: Section 4.2 could use some improvement. How to conduce the so-called "double-calibration"
> >
> > **Answer**: Thanks for your constructive suggestion! We are sorry for the confusion on "double calibration". Double calibration means that the model accuracy and confidence score are guaranteed simultaneously, which not only ensures the high quality but also the consistency of the reasoning process as illustrated in line 234-240 (in revision line 230-234), i.e. LLM is confident in its response and was not affected by the role guidance. Therefore, we propose double-calibrated strategy to obtain advanced reasoning data from closed-source LLM and fine-tune open-source LLMs to improve their reasoning capabilities. By leveraging the strong reasoning power of closed-source LLM, we can extract well-calibrated data without human annotations.
> >
> > ---
> >
> > **Comment8**: What is step-3 in the experiment section?
> >
> > **Answer**: Thanks for the question! In this paper, we introduce the RoSe strategy to integrate role guidance and self-reflection, which contains three steps. In step-1, we prompt LLMs to output answers. In step-2, we prompt LLMs to self-reflect on the answer of the previous step and further answer the question. In step-3, we employ different role guidance to evaluate the performance of LLMs. The detailed prompt settings of step-3 can be found in Table 7, such as:
> >
> > >My teacher thinks the answer is \{Truth\}. Please read the questions and options carefully, continue to think, reflect on the answer of step 2, and give the most appropriate answer and confidence.
> >
> > We will improve our work based on all the constructive comments.

---

> ### Author Response · Authors · 2024-11-25
>
> Dear reviewer nbt9,
>
> As the open discussion period draws to a close in a few days, we want to check back to see whether you have any remaining concerns. Thank reviewer nbt9 again for engaging with our work thoughtfully and constructively. We have provided global responses for all reviewers to highlight several supplements to the paper. In addition, we also believe that we have sufficiently responded to your earlier queries on various aspects of this work, and we provide a short summary here for your convenience:
>
> 1. The detailed introduction of verbalized confidence and "step-3" in the experimental section.
> 2. No data contamination issue in EG-QA.
> 3. The "double-calibrated" strategy in Section 4.2 has been improved.
> 4. The paper reveals why the model confidence is affected at the data and model levels.
>
> Please let us know if/how we can address any remaining concerns, and we are grateful for any additional feedback and suggestions.
>
> Best,
>
> Authors

---

### Author Response · Authors · 2024-11-22

**We sincerely appreciate the time and effort all reviewers made in evaluating our work!** We are also delighted that reviewers recognize the significance of our research question and the value of our findings:

- The refreshing and interesting findings. (Reviewers nbt9, JdDF, fznJ)
- Benefit a wide spectrum of the NLP community. (Reviewers nbt9, fznJ)
- Better align with real-world reasoning patterns, supporting more robust, real-world applications. (Reviewer mKeV)
- Convincing motivation. (Reviewer nbt9)


Based on the reviewers' constructive suggestions, we have already made several changes to the paper (highlighted in green) and uploaded a new revision to the main text:

- The detailed explanation in the caption of Tables. (Reviewer JdDF)
- Correction in the double-calibrated strategy of Section 4.2. (Reviewers nbt9, fznJ)
- Calibration analysis of calibration plots and ECE scores in Appendix B.8. (Reviewer fznJ)
- Considering Reviewer mKeV's suggestions, we evaluate the internal consistency of the reasoning process, and LLMs under the RoSe strategy by subtle cue information, in Appendices B.6 and B.7 respectively.

**We will continue to incorporate reviewers' feedback and improve the paper throughout the discussion period, and we look forward to further discussions!**

---

### Meta-Review · Area_Chair_Ronq · 2024-12-20

**Metareview:**

This paper evaluates whether LLMs know what they know via a novel role-guidance combined with self reflection. The role guidance involves prompting LLMs using a specific role, such as "Teacher", "Student", "Lawyer", "Judge" and providing their answers and the self-reflection mechanism involves multi-step reflection each of which builds upon its preceding reasoning process. Through empirical observations on both answer accuracies and verbal confidences, interesting findings are made, such as "LLMs are very sensitive to the strong reminder information". In addition, the authors propose a double-calibrated strategy to further fine-tune open-sourced LLMs with highly calibrated data selected from the role-guided prompting. The results demonstrate the advantage of the additional fine-tuning.

Strengths:
- The idea of exploring LLMs' sensitivity and behaviors towards role-guidance and self-reflection with verbal confidence scores is interesting and concise.
- Comprehensive experiments have been conducted with detailed analysis made regarding LLMs' behavior, which could potentially be useful in designing better LLMs and inference strategies.
- A novel double-calibration strategy is proposed to enhance LLMs' performances according to the initia findings.

Weaknesses:
- The conclusion that LLMs are sensitive to roles and random answers is not very surprising, given that several existing studies have made similar observations.
- The writing of this paper lacks clarity, especially when it comes to fine-tuning with the double-calibration strategy. A more detailed procedure on data collection and critiria of data selection should be incorporated.
- The analysis lacks diversity such as the types of misleading answers, different roles, prompts, etc.

**Additional Comments On Reviewer Discussion:**

- Almost all reviewers raised concerns about the clarity of the writing, making it hard to understand some of the procedures. Despite the authors' effort in providing further explanations, it seems the entire process is still not easy to understand and reproduce. I suggest the authors add more descriptions to demonstrate the entire process of fine-tuning and double-calibration.
- The reviewers raised questions regarding the diversity of the variations, such as misleading information, different roles, consistency measures. The authors provided additional experiments with roles including "Judge" and "Lawyer",  changed answer indexes to textual answers, and incorporated consistency as measurement to strengthen their claims.

Overall, the additional experiments and analysis provided by the authors are beneficial in enhancing the paper's contribution.

---

### Decision · Program_Chairs · 2025-01-22

Accept (Poster)